# Context-Specific Diversity of Antimicrobial Functions of Interferon-Stimulated Genes

**DOI:** 10.3390/v17121635

**Published:** 2025-12-17

**Authors:** Munesh K. Harioudh, Saumendra N. Sarkar

**Affiliations:** 1Program in Oncology, Marlene & Stewart Greenebaum Comprehensive Cancer Center, University of Maryland School of Medicine, Baltimore, MD 21201, USA; mharioudh@som.umaryland.edu; 2Department of Microbiology and Immunology, University of Maryland School of Medicine, Baltimore, MD 21201, USA

**Keywords:** interferon-stimulated genes, Interferon, oligoadenylate synthetase, virus, antiviral mechanism

## Abstract

Interferon-stimulated genes (ISGs) were initially discovered for their role in antiviral functions. However, recent studies show evidence of a diverse and context-specific regulatory function of these genes in antiviral and antibacterial protection. The molecular mechanisms of such activities vary depending on the pathogen, cell type, isoform, and species. In this review, we summarize the context-specific functions of several prominent and well-known ISG families, including OAS, IFITs, ISG15, viperin, ADAR1, and Mx proteins. We provide examples of distinct enzymatic or regulatory mechanisms that are employed by these ISGs to carry out their diverse functions, including nucleic acid sensing, RNA degradation, translation inhibition, membrane remodeling, etc.

## 1. Introduction

The early host defense against microbial infection is mediated by the cellular innate immune response, defined by the production of cytokines such as interferons (IFN), and inflammasome activation. Cytokines, through their autocrine and paracrine action, induce a set of genes in neighboring cells to provide protection, promote adaptive immune response, and finally provide tissue repair [1]. In the context of IFNs, these target genes are collectively called IFN-stimulated genes (ISGs), and due to the historic association of IFN with cellular antiviral states, they have been primarily studied in the context of virus infection [2,3]. Initially, the understanding of interferon’s action centered around its ability to induce the expression of hundreds of interferon-stimulated genes (ISGs), leading to a model often referred to as “death by a thousand cuts”. However, multiple genetic screening studies have led to an emerging paradigm that suggests that, while interferon does indeed induce a widespread transcriptional response encompassing many hundreds of ISGs, the majority of the antiviral effect against any given specific virus is context-specific and driven by a relatively small subset of these induced genes, giving rise to the “limited set” model [4]. This concept of a tailored defense system underscores the need to precisely identify which ISGs are directly responsible for inhibiting specific viruses, a critical question that has persisted in the field for nearly four decades. Furthermore, recent evidence have indicated that some of these ISGs have broader functions—including protection from other microbial infections beyond viruses [5,6]. Additionally, among over 500 ISGs, only a subset of antiviral molecular mechanisms are well-defined, while new studies continue to discover novel pathogen-specific mechanisms of action of well-known ISGs [7,8,9,10,11,12,13,14,15,16]. Here, we review context-specific—that is, pathogen- and cell-type-specific—mechanisms of action of several ISG family proteins to highlight the broader functional impact of these genes on host response.

## 2. The Landscape of Interferon-Stimulated Genes

An interferon-stimulated gene (ISG) is broadly defined as any gene whose expression is upregulated in response to interferon signaling. Interferons, upon binding to their cognate receptors on the cell surface, initiate intracellular protein signaling pathways that culminate in the transcriptional activation of a specific subset of genes involved in the innate immune system’s response to pathogens. While viral infection is a primary trigger for ISG expression, these genes can also be induced during bacterial infections and in the presence of parasites. It is estimated that a significant portion, approximately 10%, of the human genome is regulated by interferons [17]. Although the primary trigger for ISG expression is interferon signaling, it is important to note that a subset of ISGs can also be directly induced by interferon regulatory factors 3 and 7 (IRF3/7), even in the absence of downstream interferon signaling [18]. Furthermore, ISG expressions are not uniform; some ISGs exhibit basal expression levels that are further enhanced by interferon, while others show cell-type-specific expression patterns [19,20]. Interferons themselves are classified into three main types: type I, type II, and type III. While type II interferon (IFNγ) plays a crucial role in controlling intracellular pathogens and regulating tumor suppressor genes, type I (including IFNα, IFNβ, IFNω) and type III (IFNλ) interferons are the classic antiviral cytokines, triggering the expression of ISGs that combat viral infections. Notably, although type I and type III IFNs bind to different cell surface receptors, they largely signal through the same Janus kinase-signal transducer and activator of transcription (JAK-STAT) pathway, resulting in the induction of a shared array of ISGs [1,21,22,23].

## 3. ISG Responses in Distinct Cell Types

As front-line barriers, epithelia constitutively express IFNλ receptors and upregulate ISGs upon infection [23]. Influenza infection of respiratory epithelium, for example, induces ISGs, including viperin, ISG15, OASs, and Mx proteins, via the IRF3/IRF7 pathway. In epithelial cells, secreted ISG15 acts as an extracellular cytokine to signal through the LFA-1 integrin receptor on nearby immune cells, such as Natural Killer (NK) cells and T cells. This stimulation promotes the release of cytokines, most notably interferon-gamma (IFNγ). This process serves to bridge the early innate immune response initiated by epithelial cells with the broader, IFNγ-dependent response that follows [24]. In human corneal epithelium, Zika virus induced RIG-I/MDA5 and ISG expression (ISG15, OAS2, MX1) but still caused cell death—suggesting a partially effective epithelial ISG response. Whereas recombinant ISG15 added to corneal epithelial cultures directly inactivated Zika virus and prevented its entry, whereas knocking down ISG15 increased Zika infectivity [25].

Myeloid cells, such as macrophages and dendritic cells, sense pathogens and secrete type I IFN, inducing ISGs in both infected and bystander cells. In macrophages, TLR and RIG-I signaling drive strong ISG induction. For instance, HIV-1 infection of macrophages induces a distinct ISG set without inducing IFN, since HIV blocks IRF3 activation. Viperin is among the most upregulated ISGs in HIV-infected macrophages, and potently inhibits viral production in these cells [26]. Plasmacytoid dendritic cells (pDCs) also express many ISGs, but they differ functionally. Intriguingly, viperin suppresses IFN production in conventional macrophages yet enhances TLR7/9-driven IFN in pDCs. Thus, even the same ISG (viperin) can have opposite effects on IFN signaling in different myeloid subsets [27]. These cell-type differences likely arise from distinct signaling network wiring (e.g., presence of TLRs, STAT isoforms) and from interactions with other lineage-specific factors.

Neurons and glia in the CNS express IFN receptors and upregulate ISGs upon infection, but their responses are uniquely balanced to protect delicate neural tissue. In the brain, type I/III IFNs produced by infected neurons or glia induce neuronal ISGs such as viperin, OAS2, RNase L, PKR, and IFIT proteins [28]. For example, peripheral viral infections (e.g., vesicular stomatitis virus, VSV) can induce IFNβ in infected olfactory neurons, which then upregulates ISGs in uninfected CNS regions. Neurons themselves produce less inflammatory cytokines and avoid apoptosis, relying on cell-intrinsic ISGs to control infection. A notable feature is that neurons may utilize ISGs differently, e.g., the neurotropic West Nile virus is restricted by IFIT2 in neurons but not other tissues [29]. Overall, CNS ISG responses reflect cell-type-specific regulation (neurons vs. astrocytes vs. microglia) and slower IFN kinetics compared to peripheral tissues. Thus, the repertoire and magnitude of ISG activity in neurons can differ markedly from epithelial or immune cells, affecting viral tropism in the CNS. Similarly, in cancer, ISGs can play multiple and opposing roles depending on the specific gene and context. For example, ISG15 in its free form is considered anti-tumorigenic. However, when it’s conjugated to cellular proteins (ISGylation), it can promote tumor growth [30].

## 4. Context-Dependent Mechanisms of ISG Function

### 4.1. Oligoadenylate Synthetase (OAS) Family

Despite their initial characterization as antiviral ISG in humans, oligoadenylate synthetases (OAS) are ancient enzymes that polymerize ATP into 2′-5′ oligoadenylates (2-5A), and are found throughout Metazoa (from sponges to mammals) [31,32]. They belong to a broad nucleotidyltransferase (NTase) superfamily present in many cellular and viral genomes, suggesting an early origin [33]. For example, the OAS present in marine sponges can synthesize both 2′-5′ and 3′-5′ oligoadenylates and can be activated without double-stranded RNA (dsRNA), unlike vertebrate OAS [34,35,36,37]. Notably, the only known effector of OAS-produced 2′-5′ oligoadenylates (2-5A) is RNase L, which appears only in jawed vertebrates. Thus, ancient OAS homologs likely had non-IFN roles, and the modern OAS–RNase L antiviral pathway was “tinkered” together before jawed vertebrates [34]. Even in primates, loss-of-function mutations in the OAS1 genes without the corresponding compensatory mutations to increases RNase L activity argues for alternative function of OAS genes [38]. OAS proteins share a core NTase fold with cGAS; thousands of cGAS-like enzymes exist in bacteria and metazoans, underscoring an ancient lineage of sequence non-specific nucleic acid sensing NTases [39,40,41].

The human genome encodes four OAS-family genes (OAS1, OAS2, OAS3, and OASL) clustered on chromosome 12, which possibly arose by gene duplication and domain fusion. OAS1, OAS2, and OAS3 contain one, two, or three tandem OAS domains, respectively, whereas OASL (OAS-like) has one OAS domain plus two ubiquitin-like (UBL) domains at its C-terminus, indicating an ancient gene fusion event [40,42,43,44]. All four human OAS proteins are interferon-inducible and have demonstrated antiviral activity in vitro. However, extensive genetic diversity (single nucleotide polymorphisms and splice variants) exists, reflecting an evolutionary “arms race” with pathogens. For instance, a common splice-site SNP (rs10774671) in OAS1 controls production of two major isoforms: the G allele yields OAS1 P46, and the A allele yields OAS1 P42. These isoforms differ at the C-terminus: P46 has a CaaX prenylation motif (targeting it to membranes), whereas P42 lacks this motif, that has functional consequences in viral pathogenesis [45,46,47]. Similar functional consequences of membrane targeting have also been described for OAS2 [48].

#### 4.1.1. Enzyme Activity-Dependent Antiviral Function

The canonical OAS mechanism requires binding of viral double-stranded RNA (dsRNA), which activates the enzyme to polymerize ATP into 2′-5′ oligoadenylates (2-5A). These 2-5A oligomers bind and activate latent RNase L, causing RNase L dimerization and non-specific cleavage of single-stranded RNA to inhibit viral replication in a cell-intrinsic manner [49]. However, recent studies have indicated that 2-5A can be actively transported through gap junctions and can activate RNase L in a cell-extrinsic manner [50]. Thus, OAS proteins sense the presence of dsRNA and translate into an antiviral RNA-degradation response. In humans, three OAS isoenzymes (1–3) can produce 2-5A, but genetic knockout studies indicate that OAS3 is the principal enzyme required for RNase L activation. Although OAS1 and OAS2 can make 2-5A in vitro, but are dispensable for RNase L activation in certain cells [51,52]. This raises questions about the roles of OAS1/2 beyond 2-5A synthesis.

Some of these contradictory observations regarding the mechanisms of antiviral activities of OAS proteins stem from the isoform-specific function of OAS proteins. Several OAS genes contain specific SNPs at a high enough frequency that can result in different isoforms in commonly used cell lines. The OAS1 rs10774671-G allele (OAS1 P46) is strongly protective against several viruses, whereas the A allele (P42) confers higher susceptibility [45,46,53]. Mechanistically, the P46 isoform’s CaaX motif leads to prenylation and endomembrane localization, allowing it to access viral replication compartments and robustly activate RNase L. In contrast, P42 (lacking the CaaX) is diffusely cytosolic and shows poor activation in infection [45,46,47]. Consistent with this, the rs10774671-G allele (P46) is associated with reduced risk of severe COVID-19 [54,55]. Therefore, findings from OAS loss-of-function studies conducted using A/A genotype cells may not always reflect their full antiviral function.

#### 4.1.2. Enzyme Activity Independent Functions

Some OAS-family proteins have antiviral roles unrelated to 2-5A synthesis. Notably, human OASL (OAS-like) is enzymatically inactive but antiviral. Instead, OASL enhances signaling through the RNA sensor RIG-I. Its C-terminal UBL domains bind RIG-I and mimic polyubiquitin, stabilizing RIG-I in an active conformation. As a result, OASL boosts interferon induction and restrains replication of many RNA viruses in a RIG-I-dependent manner [56,57]. By contrast, mouse Oasl1 (inactive) acts as a negative regulator: it binds the 5′-UTR of the transcription factor IRF7 mRNA and suppresses its translation, thereby dampening IFN production and paradoxically promoting viral replication [58]. This mouse-specific mechanism does not occur in humans, as human OASL neither binds IRF7 mRNA nor inhibits IFN. However, in the context of DNA virus infection, a similar phenotype has been demonstrated. Human OASL and mouse Oasl2 both can bind to the DNA-sensor cGAS causing inhibition of cGAMP production resulting in the inhibition of IFN production to promote viral replication [59,60]. Another example is mouse Oas1b: this protein lacks 2-5A synthetase activity but confers resistance to West Nile virus in mice independently of RNase L. Human OAS1 P46 appears to share a similar RNase L-independent antiviral function, mediated through their ability to prolong the half-life of IFN mRNAs [47]. In sum, OAS proteins can act through diverse mechanisms—enzymatic and non-enzymatic—that are often species- and virus-specific (Figure 1).

#### 4.1.3. Other Roles: Bacterial Infection and Cancer

Beyond viruses, OAS proteins are induced by both type I and type II IFNs, which are important in antibacterial immunity. There is emerging evidence that OAS may influence bacterial infections. For example, genetic studies have linked the same rs10774671-G (OAS1-P46) allele to protection against tuberculosis, implying that OAS1 function affects intracellular bacterial control [61]. Recently, we described a unique mechanism of translational regulation of antibacterial proteins such as IRF1 and GBP1 that leads to cell-intrinsic inhibition of bacterial multiplication in vitro and in vivo [62] (Figure 2). Besides OAS1, experiments have shown that cells lacking certain other OAS proteins have higher intracellular growth of bacteria like *Listeria* and *Francisella* [49]. However, the mechanisms of antibacterial activities of other OAS isoforms are still unclear.

In cancer, chronic IFN signaling (with high ISG expression) often associates with tumor immune evasion and therapy resistance [63]. OAS genes are frequently upregulated in tumors (notably pancreatic cancer) and have been proposed as prognostic biomarkers [64,65,66,67,68]. Some studies report that reducing OASL levels can slow tumor cell growth, while others find OAS1 or OAS2 can either promote apoptosis or, conversely, promote metastasis or checkpoint blockade resistance. These seemingly contradictory roles may reflect context-dependent effects: for example, acute IFN/OAS activation can drive antitumor immunity [69,70], whereas chronic activation might upregulate immune checkpoints like PD-L1 [71,72]. Deciphering OAS functions in cancer likely requires cell-type-specific models. Preliminary data even suggest that OAS1 and OASL can sometimes promote antitumor immunity under the right circumstances. In summary, OAS proteins have multifaceted roles in immunity and disease, with functions that vary by isoform, cell type, and context.

### 4.2. The IFIT Family

Interferon-induced proteins with tetratricopeptide repeats (IFITs) are induced by type I IFN, as well as IRF3 signaling, and act as rapid effectors of the antiviral response [18,73,74]. IFIT1 specifically binds to 5′-capped RNAs lacking 2′-O methylation, blocking various steps of the translation initiation complex and ribosome recruitment. This cap-binding activity selectively inhibits translation of “non-self” viral RNAs (for example, flaviviruses, poxviruses, and coronaviruses whose 2′-O methyltransferases are mutated) [73,74,75]. IFIT1 also binds viral 5′-triphosphate RNAs, sequestering them and preventing from being translated into viral proteins [76]. IFIT2 and IFIT3 play crucial roles in regulating and enhancing the antiviral activity of IFIT1 by forming a multiprotein complex. In vitro studies show that IFIT3 and an IFIT2–IFIT3 heterodimer bind IFIT1 via a C-terminal motif, stabilizing IFIT1 and enhancing its binding to cap0 RNAs. Accordingly, IFIT3 expression augments IFIT1-mediated translation inhibition (e.g., on a Zika virus reporter) and is critical for restricting viruses lacking 2′-O caps [77]. Thus, IFIT1 alone recognizes cap0 RNA, while complexes of IFIT1/2/3 are most inhibitory. Notably, human IFIT2/3 also promote innate signaling (e.g., IFIT3 potentiates RIG-I/MAVS signaling) [78,79].

The three IFITs have isoform- and context-specific roles. IFIT1 is broadly antiviral in cells with high IFIT expression (e.g., epithelia or fibroblasts) [80]. IFIT2 is particularly important in the central nervous system: in mice, IFIT2 is highly induced in neurons and glia during neurotropic infections and is essential to curb viral spread in the brain [81,82]. For example, *Ifit*^−/−^ mice succumb to vesicular stomatitis virus (VSV) or mouse hepatitis virus (MHV) encephalitis with massive neuronal infection, whereas IFIT1 plays little role in that setting. Conversely, in myeloid cells (macrophages, dendritic cells), IFIT1 often dominates the antiviral program. IFIT2 (and IFIT3) are also expressed in macrophages but appear to modulate inflammatory signaling. For instance, IFIT2 enhances IRF3 phosphorylation and cytokine production in infected macrophages, contributing to both pathogen restriction and immune activation [83]. In contrast to antiviral activity, recent studies showed that IFIT2 and IFIT3 can promote influenza A virus infection by binding to viral and cellular messenger RNAs, enhancing translational efficiency and, ultimately, viral replication [84,85].

### 4.3. ISG15

ISG15 (Interferon-Stimulated Gene 15) is a ubiquitin-like (Ubl) protein (~15–17 kDa) that is one of the most strongly induced genes in response to type I interferons (IFNs) [86,87]. Upon IFN induction, ISG15 is activated by a dedicated enzymatic cascade analogous to ubiquitination. The E1 activating enzyme UBE1L (UBA7) forms a thioester with ISG15, which is then transferred to the E2 conjugating enzyme UbcH8 (UBE2L6). In human cells, the principal E3 ligase is HERC5; intriguingly, mice lack a true HERC5 ortholog and instead use HERC6 for ISGylation. These enzymes are all IFN-inducible, ensuring that ISGylation activity parallels ISG15 expression. The C-terminal LRLRGG motif of ISG15 is essential for conjugation, forming an isopeptide linkage to target lysines. Conjugated ISG15 can be removed by the protease USP18 (also called UBP43), which is the major ISG15-specific deubiquitinase. Thus, ISGylation is reversible, providing dynamic control of protein modification during infection [24,86,88].

Human and mouse ISG15 genes encode homologous proteins but with striking functional divergences. Human and mouse ISG15 share only ~65% sequence identity, with critical differences affecting binding partners. The most notable disparity is in IFN regulation: human free ISG15 binds strongly to USP18 and stabilizes it by preventing its proteasomal degradation. This enables USP18 to negatively regulate type I interferon (IFNα/β) signaling by binding to IFNAR2 and inhibiting the activation of JAK1 and TYK2 [89,90]. Mouse ISG15, by contrast, binds USP18 weakly and does not appreciably affect USP18 stability; consequently, *Isg15*^−/−^ mice do not display IFN hyperactivity. These molecular differences have dramatic physiological consequences. ISG15-deficient humans suffer from type I interferonopathy and mycobacterial disease (due to loss of IFN-γ induction) but show no unusual susceptibility to viral infections. In stark contrast, *Isg15*^−/−^ mice exhibit high vulnerability to diverse viral infections, succumbing rapidly to viruses that wild-type mice resist [91]. For example, *Isg15*^−/−^ mice have greatly increased mortality after influenza A or B challenge, demonstrating ISG15′s antiviral role in rodents [92]. Other mechanistic differences include the E3 ligase usage: human cells employ HERC5 to mediate ISGylation, whereas mice rely on HERC6 (mouse HERC5 is inactive). In summary, while human ISG15 is highly tuned to regulate cytokine responses and proteostasis via USP18, mouse ISG15 functions more as a direct antiviral effector—a distinction that must be kept in mind when extrapolating mouse studies to human biology.

### 4.4. Viperin (RSAD2)

Viperin (virus inhibitory protein, endoplasmic reticulum-associated, interferon-inducible; gene RSAD2) is an ancient radical SAM enzyme induced by interferons. It comprises an N-terminal amphipathic α-helix that anchors the protein to the cytosolic face of the endoplasmic reticulum (ER). A central radical S-adenosylmethionine (SAM) domain binds a [4Fe-4S] cluster via an almost invariant RS motif, CX_3_CX_2_C, while a C-terminal extension completes the substrate-binding pocket. A conserved C-terminal tryptophan motif is required for recruitment of the cytosolic iron–sulfur assembly component (CIAO1) to install the Fe-S cluster. In human cells, viperin is constitutively present at very low levels in some tissues, but it is robustly upregulated by cytokines and pathogen signals. In particular, type I and III interferons (IFN-α/β/λ) strongly induce RSAD2, making it a canonical interferon-stimulated gene. Viral pattern-recognition receptor (PRR) pathways (e.g., RIG-I, MDA5, TLRs) capable of activating IRF3/7 and NF-κB, can also bind the RSAD2 promoter and activate its transcriptional induction [93,94].

Since its discovery as a gene capable of inhibiting human cytomegalovirus (HCMV), viperin (also known as cig5) has been one of the most studied ISGs due to its potent and broad-spectrum antiviral activities across species and cell types. These studies have suggested multiple and sometimes contradictory mechanisms of inhibition for different viruses. Some of these mechanisms include perturbation of lipid metabolism in the context of Influenza A virus infection [95], protein–protein interactions for the inhibition of hepatitis C and Zika Virus infection [96]. However, the discovery of its antiviral ribonucleoside synthase activity, along with evidence from molecular evolution studies, has indicated that, similar to several other ISG, viperin has an ancient origin, and its enzyme activity contributes to a broader antiviral function across the tree of life [97,98,99]. Biochemically, viperin is a radical SAM enzyme: it cleaves SAM to generate a 5′-deoxyadenosyl radical and catalyzes the dehydration of CTP. The enzyme converts cytidine triphosphate into the unnatural nucleotide 3′-deoxy-3′,4′-didehydro-CTP (ddhCTP) via a radical-mediated mechanism. In mammalian cells, expression of viperin (or IFNα treatment) leads to accumulation of ddhCTP. This compound lacks the 3′-hydroxyl of the ribose, so when ddhCTP is incorporated by viral RNA-dependent RNA polymerases it terminates chain elongation. Indeed, ddhCTP acts as a natural chain terminator for flavivirus polymerases and potently inhibits Zika virus replication in vivo. In short, viperin’s core activity is the radical-mediated transformation of CTP into a novel antiviral nucleotide (ddhCTP), providing a partially unifying mechanism for its broad antiviral effects [93]. Related viperin-like enzymes in bacteria and fungi use similar SAM chemistry to modify other substrates, underscoring the enzyme’s ancient origin [99]. Beyond viruses, evidence exists for the antibacterial function of viperin as well [100]. Taken together, these findings again emphasize the mechanistic specialization of another ISG—viperin in a context-specific manner.

### 4.5. ADAR1

Adenosine deaminase acting on RNA 1 (ADAR1) is an interferon-stimulated gene that encodes an RNA-editing enzyme. ADAR1 catalyzes the deamination of adenosine to inosine (A-to-I) in double-stranded RNA (dsRNA) [101]. This process can have diverse effects on viral replication and the host immune response. ADAR1 exists in two main isoforms: a constitutively expressed p110 form primarily located in the nucleus, and an interferon-inducible p150 form found in both the cytoplasm and the nucleus. Both isoforms share a catalytic deaminase and double-stranded RNA-binding domains [102].

ADAR1’s primary physiological role is to prevent autoinflammation induced by self-RNA recognition. In the absence of ADAR1 editing, cellular dsRNAs (often from Alu/(TA)_n_ repeats or misfolded transcripts) activate cytosolic sensors such as MDA5. Indeed, *Adar1*^−/−^ mice die mid-gestation with a massive interferonopathy that is fully rescued by deleting MDA5 (or MAVS) [103,104,105]. This indicates that ADAR1 normally antagonizes the MDA5–MAVS pathway via A-to-I editing. In human patients, ADAR1 mutations (often affecting p150’s Zα domain) cause Aicardi–Goutières syndrome (a type I interferon autoinflammatory disorder), underscoring ADAR1’s role in self vs. non-self discrimination. Beyond editing, ADAR1 p150 binds to Z-RNA, left-handed dsRNA, produced during viral replication and endogenous cellular processes (such as oxidative stress), and prevents activation of pathogenic type I responses by PKR and ZBP1 [11,106,107]. Thus, ADAR1 (especially p150) acts as a gatekeeper: it edits self dsRNA so that MDA5 stays quiescent, and it dampens PKR/ZBP1 to prevent excessive cell death. However, in neurons, an evolutionarily conserved function of ADAR1 and ADAR2 together is to edit key transcripts for brain function (e.g., glutamate receptor recoding) [108]. In fact, ADAR-mediated RNA editing has been found to extensively recode the neural proteome in octopus [109]. ADAR1 loss in neurons may cause cell stress or trigger immune responses that contribute to neurologic disease.

In the context of virus infection, ADAR1 impact is highly variable dependent on the virus. Many viruses benefit from ADAR1 activity. For example, during influenza infection, ADAR1 (particularly p150) suppresses antiviral signaling, thereby promoting replication. Vogel et al. showed that deleting ADAR1p150 in human cells led to sustained RIG-I–IFN activation and cell death upon influenza infection, which strongly inhibited viral growth. In contrast, removing the nuclear p110 isoform made cells more permissive to influenza, suggesting p110 has some opposing, virus-restrictive effect [110]. Measles virus and other negative-strand viruses also exploit ADAR1 to evade immune detection. Thus, ADAR1 often has proviral roles by editing viral or cellular RNAs to keep RLR pathways in check. Conversely, ADAR1 editing can also be antiviral in some cases. Hyperediting of viral genomes (e.g., hepatitis C virus, LCMV) can destabilize viral RNAs and inhibit replication [101].

### 4.6. Mx Proteins

Mx proteins, including MxA and MxB in humans, are large dynamin-like GTPases that represent a key family of interferon-induced antiviral effectors [111,112]. Their expression is primarily induced by type I and type III interferons [112,113]. These proteins are evolutionarily conserved across vertebrates and are known for their broad antiviral activity against a variety of RNA and some DNA viruses. Their homology to dynamin, a protein involved in membrane remodeling processes, suggests a potential mechanism involving mechanical disruption of viral structures or interference with membrane-associated events in the viral life cycle [114].

Human MxA is a cytoplasmic protein with a broad antiviral activity, inhibiting the replication of diverse viruses, including influenza A virus, vesicular stomatitis virus, measles virus, and hepatitis B virus. MxA is thought to act as an intracellular barrier, often by targeting viral ribonucleoproteins (RNPs) and inhibiting their function [113]. For instance, against influenza A virus, MxA can retain the incoming viral genome in the cytoplasm, preventing its transport to the nucleus, a step crucial for viral replication. A specific region within MxA, known as loop L4, has been identified as a key determinant of its antiviral specificity, particularly against orthomyxoviruses like influenza A and Thogoto virus [115,116,117]. However, MxA does not appear to exhibit significant antiviral activity against certain poxviruses like vaccinia virus.

Human MxB, in contrast to MxA, is primarily associated with the outer membrane of the cell nucleus and exhibits antiviral activity against a different set of viruses, notably retroviruses such as HIV-1 and herpesviruses like HSV-1. Against herpesviruses, MxB has been shown to interfere with an early step of replication, affecting alpha-, beta-, and gammaherpesviruses by disassembling incoming viral capsids, thus hindering the delivery of the viral genome into the nucleus. For HIV-1, MxB inhibits viral replication by blocking the nuclear import of the viral preintegration complex [118,119]. Interestingly, MxB has also been reported to inhibit hepatitis C virus by binding to the viral NS5A protein and impairing its interaction with cyclophilin A. The distinct subcellular localization and viral specificities of MxA and MxB highlight the specialized roles of these two closely related proteins in the interferon-mediated antiviral defense [120,121].

### 4.7. IRFs and Nucleic Acid Sensors

Beyond the above-mentioned classic ISGs, several IRF-family and STAT-family transcription factors are also induced by IFNs and function in a context-specific manner. In multiple cell types, constitutively expressed IRF3 functions as the transcription factor for the initial induction of IFNβ, whereas IRF7 is induced later through autocrine IFN signaling to further promote and sustain IFN induction [122]. However, the majority of type I IFN, particularly IFNα subtypes, are produced by the plasmacytoid DC, which uses cell-type specific constitutive expression of IRF7 as the transcription factor for IFN induction [123]. Although IRF family members are best studied for their critical involvement in type I IFN gene induction, immune cell-type-specific functions of multiple members such as IRF4 and IRF8 have been recognized (reviewed in [122]). In the context of oncogenesis, multiple IRFs have been noted to have opposing roles in a tumor or immune cell-specific manner. Gene knockout and other studies have implicated tumor suppressor functions of IRF1, IRF3, and IRF7, which have been attributed to their roles in influencing T cell priming and function [124,125]. However, the same molecules can also affect tumor growth and immune evasion of the tumor cells [71]. In a similar manner, the ability of inducing IRF1 by IFNβ seems to be critical for proinflammatory response of IFNβ rather than IFNλ in a tissue-specific manner [126]. IRF1 also promotes cell-intrinsic protection from bacterial growth by promoting expression of guanylate-binding proteins (GBPs) [127,128].

Expression of multiple nucleic acid sensors such as RIG-I, cGAS, and a few TLRs is also increased by exposure to IFN, which are utilized in a cell-type-specific manner. RIG-I seems to be necessary during WNV infection to provide a protective IFN response in the periphery [129]. In contrast, TLR3 seems to be important for protection in the CNS [130,131]. Similarly, cGAS and TLR3 are differentially needed in a tissue-type-specific manner to protect from severe pathology associated with HSV infection [132,133,134]. Another example of cell-type-specific utilization of nucleic acid sensors is observed in pDC. Although RLR or cGAS are the predominant sensors for IFN induction in the majority of cell types, pDC seems to primarily utilize TLRs (TLR9 or TLR7/8) for IFN induction.

## 5. Conclusions

Induction of the antiviral state following IFN stimulation has long been ascribed to the collective antiviral actions of ISGs. However, as discussed above, recent studies have shown that ISG functions vary across pathogens, tissues, developmental stages, and species. The examples discussed in this review—OAS proteins, IFITs, ISG15, viperin, ADAR1, and Mx proteins—illustrate a recurring theme: each ISG family comprises a collection of molecular strategies shaped by evolutionary pressures and fine-tuned to specific cellular contexts. These mechanisms span enzymatic activities such as oligoadenylate synthesis and RNA modification, direct recognition of non-self molecular patterns, modulation of innate signaling pathways, and interactions with host or pathogen proteins (Figure 3).

Increasingly, studies reveal that ISGs extend their functions beyond classical antiviral roles to influence antibacterial immunity, inflammation, and cancer biology. This broader view highlights ISGs not as static antiviral factors, but as dynamic regulators integrated into the larger framework of host defense and immune homeostasis. Importantly, genetic polymorphisms, alternative splicing, cell-type specialization, and species-specific differences can fundamentally alter ISG function, underscoring the need for careful interpretation of experimental systems. As new genetic and biochemical tools continue to uncover previously known functions of ISGs, future research will likely reveal an even more complex network of pathogen- and context-dependent activities. Understanding these layers of regulation will be critical for leveraging ISGs as therapeutic targets or biomarkers in infectious disease, autoimmunity, and cancer. Ultimately, dissecting the mechanistic diversity of ISGs provides a powerful window into how the innate immune system evolves, adapts, and orchestrates precise responses to an ever-changing array of microbial threats.

## Figures and Tables

**Figure 1 viruses-17-01635-f001:**
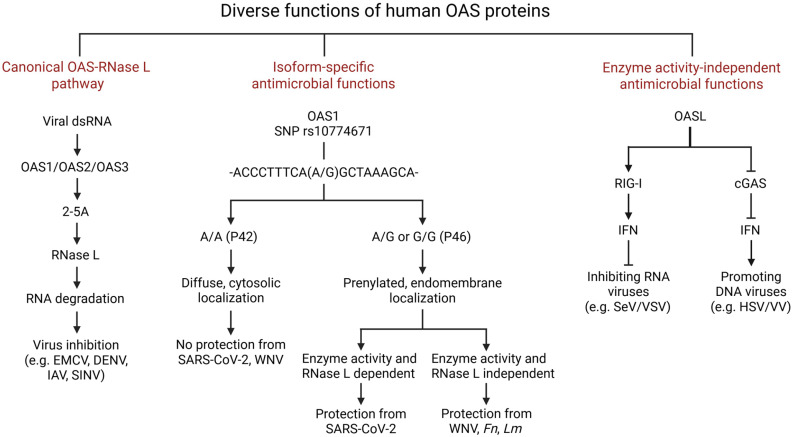
Diverse functions of human OAS proteins. The human OAS family consists of 4 genes—OAS1, OAS2, OAS3, and OASL. Except for OASL, all other OAS proteins are enzymatically active. Canonically, after getting induced by IFNs, OAS proteins bind viral dsRNA and are activated to synthesize short 2′-5′ oligoadenylates (2-5A) from ATP. These 2-5A molecules bind and activate RNase L, which degrades cellular and viral RNA to inhibit translation. Isoform-specific functions of OAS proteins are illustrated, specifically for OAS1. The OAS1 SNP rs10774671 in its C-terminal region causes alternative splicing, resulting in two isoforms: P42 and P46, originating from allele A and G, respectively. This OAS1 SNP influences susceptibility to viral (such as SARS-CoV-2 and WNV) and Mtb infections. Prenylated OAS1 P46 localizes to the endomembrane region and is able to inhibit viral replication by binding to viral RNA (SARS-CoV-2) or by protecting IFNβ mRNA from degradation (WNV). Human OASL is enzymatically inactive but plays a dual role against viruses: it inhibits RNA viruses by binding to and enhancing RIG-I-mediated signaling, leading to increased type I interferon production, which suppresses viral replication. However, it promotes DNA virus infection through binding to cGAS. This binding inhibits cGAS enzyme activity and limits interferon production, thereby enhancing DNA virus replication. EMCV, Encephalomyocarditis virus; DENV, Dengue virus; IAV, Influenza A Virus; SINV, Sinbis virus; WNV, West Nile virus; *Fn*, *Francisella novicida*; *Lm*, *Listeria monocytogenes*; SeV, Sendai virus; VSV, Vesicular stomatitis virus; HSV, Herpes simplex virus; VV, vaccinia virus.

**Figure 2 viruses-17-01635-f002:**
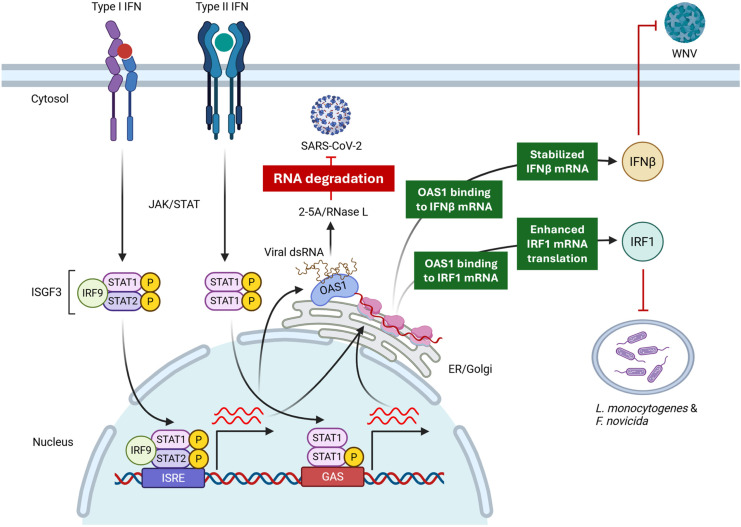
Context-specific mechanisms of antimicrobial activities of OAS1. OAS1, through its canonical enzyme activity via RNase L, inhibits SARS-CoV-2 through RNA degradation. On the other hand, OAS1 inhibits WNV through a non-canonical mechanism by sequestering and protecting IFNβ mRNA from degradation. This leads to the upregulation of IFNβ, which, through IFNAR signaling, inhibits WNV. In case of bacterial infection, OAS1 inhibits intracellular bacteria through an NTase- and RNase L-independent mechanism by enhancing translation of IRF1 mRNA.

**Figure 3 viruses-17-01635-f003:**
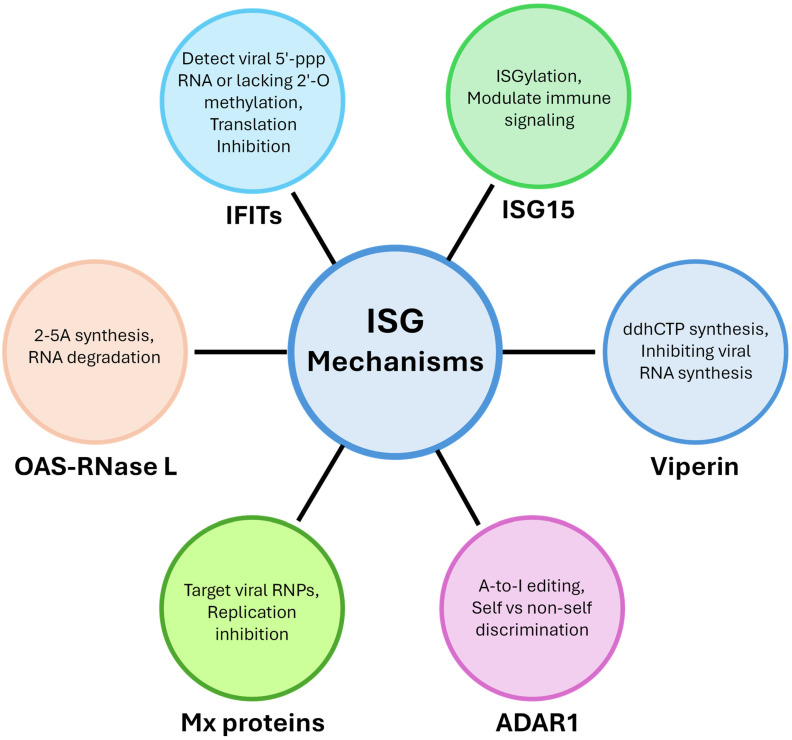
Mechanisms of ISGs. ISGs combat infection through diverse mechanisms, including RNA degradation by the OAS-RNase L pathway, translation inhibition by IFIT proteins, modulation of immune signaling through ISGylation by ISG15, inhibition of viral RNA synthesis by viperin, discrimination of self vs. non-self through A-to-I editing by ADAR1, and replication inhibition by Mx protein through targeting viral RNPs.

## Data Availability

No new data were created or analyzed in this study.

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
