# Peer review of "Context-Specific Diversity of Antimicrobial Functions of Interferon-Stimulated Genes"

_viruses, 2025, doi:10.3390/v17121635_

Round 1
Reviewer 1 Report
Comments and Suggestions for Authors
This is an interesting review that I enjoyed reading. It is nicely written and provides a great overview of the context-specific functions of ISGs. This review focuses mostly on a small subset of ISG but this does not constitute a limitation as these ISGs are maybe the most important ISGs and the most described one.
My only criticism about this review is that it is lacking schematics that try to summarize what the text is describing. There is a lot of information in each section of the review describing how in different cell types, during different virus infection or between mouse and human, different ISGs have distinct functions. After reading the review the reader might feel overwhelmed and struggle to precisely remember an important characteristic of a given ISG. Accompanying schematics for each ISG summarizing the context-specific function will help the reader find back easily precise information. This will really improve this review and make this manuscript a reference review for the field.
- Line 38: please clarify "death by a thousand cuts". I am not sure that this expression is known by non-native speaker.
- Line 64: I suggest adding the following review that nicely summarizes how ISG can be induced in an IFN-independent manner: https://journals.asm.org/doi/10.1128/mbio.02459-24
- A schematic depicting the various outcomes of OAS-mediated signaling (depending on the isoform) will help the reader. Same for OAS1.
- For the part describing the role of ISG15 through its interaction with USP18: while the authors mentioned the role of USP18 in removing ISG15 from protein, I do think (unless I missed it) that the authors forgot to mention the role of USP18 in negatively regulating IFN signaling. This will help clarify why interaction of free ISG15 with USP18 leads to negative regulation of IFN signaling by stabilizing USP18.
- Line 315: extra "." after "in"
- The only figure in this review does not seem to be listed in the text.
- The conclusion section of this review falls a little flat and also does not focus so much on the context-specific expression and functions of ISG. It would be great if the authors could use the conclusion section to speculate on the origin or the context-specific function of the ISG they described in the work. The field might not fully know why, but there should be studies speculating on the proposing driving mechanisms. The authors could also speculate on mechanism-like epigenetic differences, unique signal transduction pathways etc...
Reviewer 2 Report
Comments and Suggestions for Authors
Please find enclosed my review by Hariodh and Sarkar. This review focuses on the important class of immune defense and immunomodulatory proteins known as interferon-stimulated genes (ISGs). Historically, ISGs have largely been characterized in light of their antiviral functions and often viewed as one-dimensional in terms of their activity. It has become increasingly appreciated that the individual activities are influenced by a number of factors indicating how the context influences outcomes. Several of these contexts, which are the focus of this review, include cell type and isoform specific functions. The review focuses on some of the classic ISGs (OAS, IFIT, ISG15, etc) as examples.
As many still view ISG factors as one-dimensional, I think the review is timely as there is a need for a cohesive review/report highlighting these contexts-specific activities which are likely the norm and not the exception. The review will do much to showcase that.
I have no major critiques just some suggestions, most of which are minor.
I think this is very nice topic to review and will be impactful.
Primary suggestions
1) the citations need to be double-checked as they seem shifted in many cases throughout.
Examples:
Line 120, ref 28 for an OAS sentence…ref 28 is for the ISG15 cancer paper…should be ref 29 Hartmann OAS paper
Line 154, ref 43 for 2-5A transport…is a review on OAS, where I think the reference should be ref 44, by Huai in Immunity
There are others, so please review.
2) consider adding to relevant contexts: 1) differential induction across cell types, as well as constitutive expression, 2) timing, 3) experimental items, cell culture vs. in vivo, many ISGs have been worked out in cell culture (differences in monoculture, metabolism, oxygen tension among other things), 4) in addition, in some cases the differences in activity actually might be due to misannotation and comparisons between paralogs not orthologs (IFITs; https://pubmed.ncbi.nlm.nih.gov/27240734/).
3) I think it is valuable to potentially further emphasize repertoire/composition of the ISGs, that there is more crosstalk than appreciated, and differential expression across cell types impacts these context-specific activities.
4) if possible, consider, adding another figure and/or table either listing general principles or diagramming these things like 1) cell type specificity, 2) isoforms, 3) both antiviral and antibacterial, 4) proviral ISGs vs. some viruses, antiviral vs. others, 5) repertoire, 6) constitutive expression (e.g., ESCs), 7) timing. Could list the relevant ISGs highlighted herein under the different categories. Could be visualized as cell, boxes, a circle, among other ways.
Adding a summary figure beyond the OAS figure already present will add to value of the review, probably increase citations, used by folks in presentations, and catch the eye of folks skimming the literature.
5) small, and probably editorial, if the formatted version is near final, figure could be moved forward in the text instead of last page.
Minor
1) Line 66: basal expression of ISGs, could use citations:
https://pubmed.ncbi.nlm.nih.gov/29249360/
https://pubmed.ncbi.nlm.nih.gov/39475961/
https://pubmed.ncbi.nlm.nih.gov/30988429/
there are also likely other relevant papers
2) line 121, OAS-encoded by viral genomes, consider citing the squirrelpox OAS paper https://pubmed.ncbi.nlm.nih.gov/24983354/
And this paper for ancient https://pubmed.ncbi.nlm.nih.gov/19904482/
3) line 130, thousands of bacteria like cgas proteins https://www.nature.com/articles/s41586-019-0953-5
4) line 321…ADAR, what are Ta repeats…is it TATATATATA…if so maybe capitalize the “A” and do (TA)n, if not clarify with a few words
5) ADAR, maybe consider include mentioning the Z-RNA binding domain…as Z-RNA is a PAMP…and involved in host defense pathways like necroptosis, and viruses encoded Z-RNA binding domains
6) MX, maybe consider citing some of the original/early studies identifying the resistance and the protein, same goes for some of the factors noted
7) MX, I these were missed https://pubmed.ncbi.nlm.nih.gov/24121441/
https://pubmed.ncbi.nlm.nih.gov/24055605/
8) IRF1, these studies should be considered as they illustrate the contexts of constitutive expression and metabolic state impact IRF1-mediated antiviral activities.
https://pubmed.ncbi.nlm.nih.gov/39475961/
https://pubmed.ncbi.nlm.nih.gov/30988429/
9) Fig.1, consider adding the OAS protein binding and/or sequestering IFNB mRNA as an intermediate step before RNA stability
10) Fig. 1, consider, for the translation part, adding an intermediate step displaying the OAS enzymatic independent functions
11) note proviral ISG activities…IFITs, https://pubmed.ncbi.nlm.nih.gov/32839537/
https://pubmed.ncbi.nlm.nih.gov/40497724/
